# Future trends of marine fish biomass distributions from the North Sea to the Barents Sea

Cesc Gordó-Vilaseca [1] ✉, Mark John Costello [1], Marta Coll [2,3], Alexander Jüterbock[1], Henning Reiss[1] & Fabrice Stephenson[4,5]

Climate warming is one of the facets of anthropogenic global change predicted to increase in the future, its magnitude depending on present-day decisions. The north Atlantic and Arctic Oceans are already undergoing community changes, with warmer-water species expanding northwards, and colder-water species retracting. However, the future extent and implications of these shifts remain unclear. Here, we fitted a joint species distribution model to occurrence data of 107, and biomass data of 61 marine fish species from 16,345 fishery independent trawls sampled between 2004 and 2022 in the northeast Atlantic Ocean, including the Barents Sea. We project overall increases in richness and declines in relative dominance in the community, and generalised increases in species' ranges and biomass across three different future scenarios in 2050 and 2100. The projected decline of capelin and the practical extirpation of polar cod from the system, the two most abundant species in the Barents Sea, drove an overall reduction in fish biomass at Arctic latitudes that is not replaced by expanding species. Furthermore, our projections suggest that Arctic demersal fish will be at high risk of extinction by the end of the century if no climate refugia is available at eastern latitudes.

Climate warming is driving changes in the distribution of many species. Expanding ranges towards higher latitudes and contracting ranges in lower latitudes have been widely reported, and are resulting in species richness shifts[1–3]. These distributional shifts are driven by local climate velocities, which often differ from place to place, and do not strictly follow the global patterns of temperature, both in direction and magnitude of change[4]. The Arctic, warming almost four times faster than the global average[5], is experiencing increases of species richness due to the expansion of several warmer-water species, and the contraction of fewer colder-water species; and these changes are expected to continue in the future[6–8].

Marine fish distribution shifts have significant implications for ecosystems and human activities, particularly for the fishing industry[9–11], and could result in transboundary conflicts due to the redistribution of commercially important fish species worldwide[12–14]. For all fish species, including non-commercial species, conservation efforts may be challenged by the climate-induced displacement of populations from marine protected areas, or by ecosystem-wide changes derived from species geographic range shifts effects on species interactions, predator-prey dynamics, or food webs[15–17]. In the last decades, SDM projections into the future have provided relevant insights to policy makers, fisheries and conservation managers[18–20]. However, future projections of fish distributions to date either (1) do not include species' relative abundance or biomass, (2) model species independently, and/or (3) focus on few, common species, often limited to such of interest for fisheries with sufficient data to run species distribution models (SDM).

To overcome these limitations, we applied a joint species distribution model (J-SDM) of 107 Northeast Atlantic marine fish

[1]Faculty of Biosciences and Aquaculture, Nord University, Bodø, Norway. [2]Institute of Marine Sciences (ICM-CSIC), Barcelona, Spain. [3]Ecopath International Initiative (EII), Barcelona, Spain. [4]School of Natural and Environment Sciences, Newcastle University, Newcastle upon Tyne, UK. [5]School of Science, University of Waikato, Hamilton, New Zealand. ✉e-mail: cescgordo@posteo.eu

distributions along the continental shelf from the North Sea to the Barents Sea (61 of them including their biomass distribution). We analysed richness and relative dominance trends in Arctic communities with potential changes in species distributions and biomass (as relative abundance) under different future scenarios and investigated the influence of species traits on these future distributions.

The recent development of J-SDMs, and particularly of the Hierarchical Modelling of Species Communities (HMSC) framework, represents an advance over traditional SDMs, and is able to partially overcome the listed limitations by assuming a joint response of species to the environment and to each other[21,22]. This allows rare species to «borrow» niche information from more common species, particularly from those closely related phylogenetically. Moreover, JSDMs account for co-occurrence patterns betwen species by using latent variables. Although the interpretation of these correlation patterns into biotic interactions cannot be made easily[23,24], accounting for them may provide better estimation of the environmental parameters of the model[25]. For all this, JSDM-HMSC has proven to be among the best predictive statistical distribution models for species communities, particularly in the presence of several rare species[22,25].

In the Northeast Atlantic and Arctic Barents Sea, rising temperatures have already led to altered ocean circulation patterns, a decrease in sea ice cover, and profound changes to the marine ecosystems[26]. In the North Sea, widespread northward displacements have been documented in the planktonic community, in the pelagic and demersal fish communities, and are expected in benthic communities[27–29]. Similarly, the Barents Sea has experienced an arrival of boreal species and a decline of Arctic species, which have their trailing edge within the Barents Sea[30,31], leading to compositional changes in the Barents Sea communities, including increases in species richness[30,32]. Moreover, Arctic species are shifting their biomass centroids northwards at a higher rate than boreal species[8]. Although these changes are expected to continue in the future[6,7], their extent, and their implications for the biomass of future communities remain little investigated. For example, it is unclear to what degree Arctic species will retract or fully disappear from the Barents Sea with climate warming. Understanding the implications of expected range shifts is of critical

significance for Arctic communities, given the Arctic's accelerated warming, the associated higher extinction risk of polar species, and the inherent limitation of Arctic demersal fishes to shift into northern latitudes due to the absence of a contiguous continental shelf[5,33,34].

Here we present a JSDM model of the boreal and arctic marine fish communities from the North Sea to the Barents Sea (Fig. 1), and we show overall increases in richness and declines in relative dominance in the community with projected future conditions, as well as generalised increases in species' ranges and abundance. This comes at the cost of severe declines of Arctic species. Furthermore, the practical disappearance of the two most common fish species in the Barents Sea, namely capelin and polar cod, results in an overall reduction in fish biomass. We predict that Arctic demersal fish species will be at high risk of extinction in the next decades if no climate refugia is available at eastern latitudes.

## Results

### Environmental correlations

Among selected environmental variables, depth was the best predictive variable in both the presence-absence and biomass models, with an average of 58% and 49% of deviance explained, respectively, followed by sea bottom temperature with 32% and 19% respectively, and the spatial random effect, which accounted for 8% and 29% respectively. Each of the other variables explained on average less than 1% of the total deviance in both models, although this varied by species (Supplementary Fig. 1). For example, phytoplankton concentration explained 20% of variance in the thorny skate (*Amblyraja radiata*) probability of occurrence, and sea ice concentration explained 15% of variance in herring (*Clupea harengus*) CPUE distribution.

We found strong support for phylogenetic niche conservatism, with a phylogenetic correlation parameter *rho* of 0.58, 95% CI [0.41,0.73] in the presence-absence model, and 0.96, 95% CI [0.89,0.95] in the CPUE model, which strongly suggests that species niches in the community are highly determined by phylogenetically structured traits. After accounting for the fixed effects, representing species responses to the environment conditioned on their traits, we found pronounced residual species co-occurrence patterns with

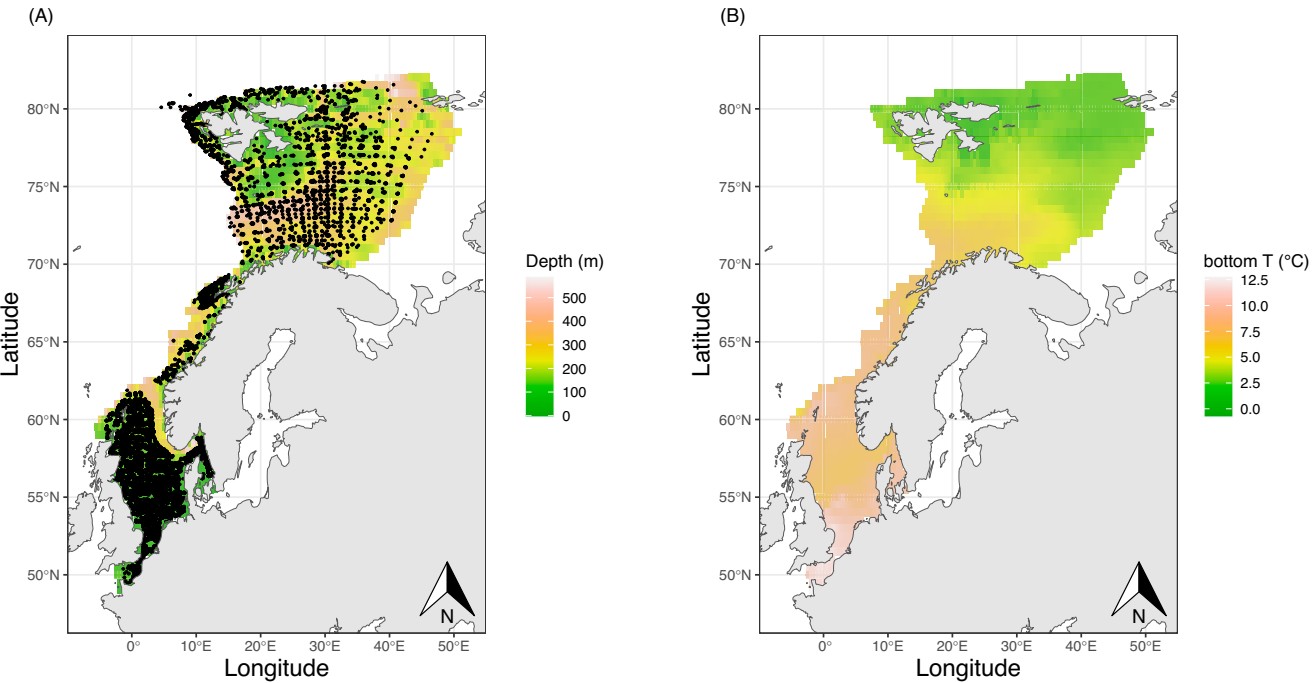

**Fig. 1 | Study area.** Black dots correspond to trawls between 2004-2022, background raster shows (**A**) mean depth and (**B**) annual mean bottom temperature in each cell at a resolution of 0.25° × 0.25°.

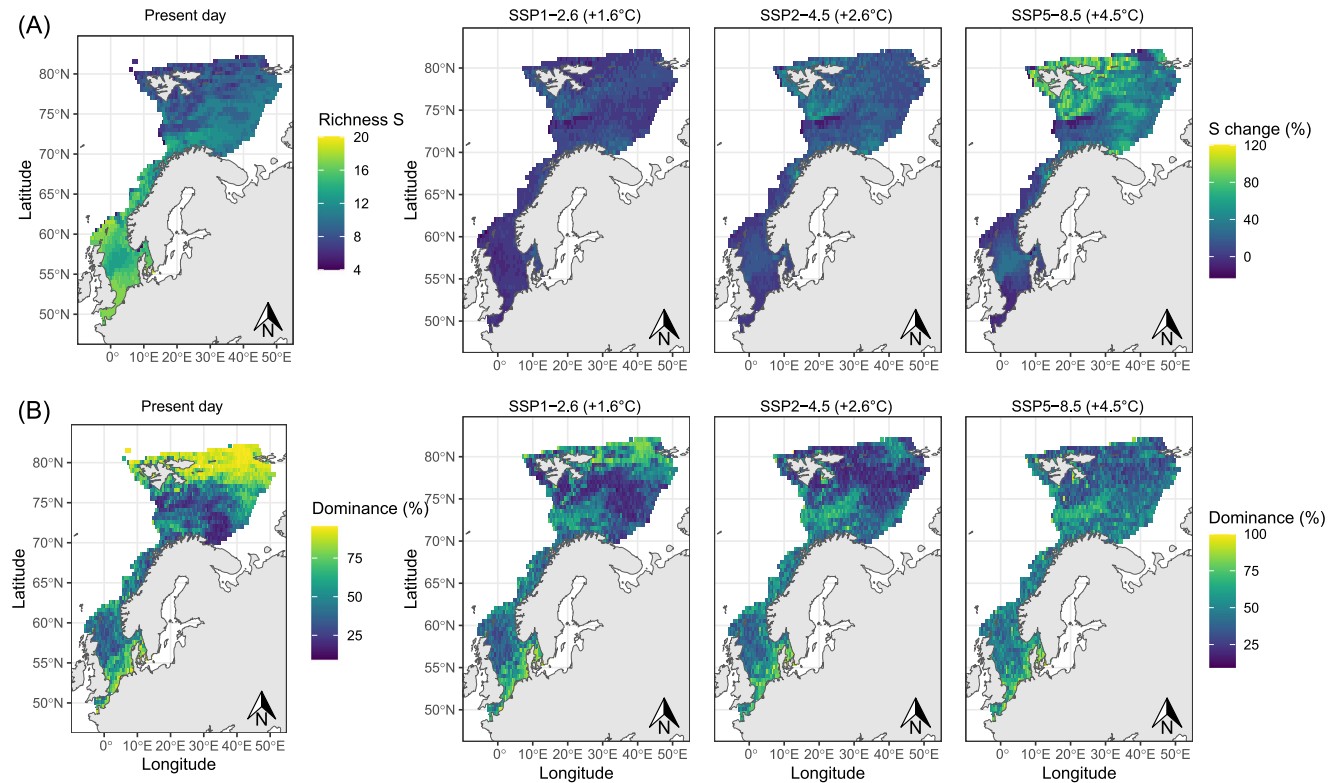

**Fig. 2 | Richness and dominance distribution change.** Projected alpha diversity changes in **A** species richness and **B** relative dominance as percentage of the highest species' biomass to total species' biomass in present day and 2100 under different shared socioeconomic pathways.

strong statistical support ($p < 0.05$, Supplementary Figs. 2 and 3), although without obvious community patterns.

### Species richness and relative dominance

Species richness was projected to increase in the study area across future scenarios (Fig. 2). The biggest increases were projected in the

### Table 1 | Share of area of each of the dominant species (species with highest biomass at each cell)

| Species | Present | SSP1-2.6 (+1.6 °C) | SSP2-4.5 (+2.6 °C) | SSP5-8.5 (+4.5 °C) |
|---|---|---|---|---|
| *Boreogadus saida* | 29 | 15 | 5 | 0 |
| *Eutrigla gurnardus* | 0 | 1 | 1 | 0 |
| *Gadus morhua* | 3 | 4 | 7 | 0 |
| *Hippoglossoides platessoides* | 4 | 6 | 5 | 0 |
| *Limanda limanda* | 24 | 25 | 25 | 23 |
| *Mallotus villosus* | 1 | 0 | 0 | 0 |
| *Melanogrammus aeglefinus* | 0 | 0 | 0 | 2 |
| *Merlangius merlangus* | 15 | 16 | 18 | 21 |
| *Micromesistius poutassou* | 12 | 18 | 21 | 21 |
| *Reinhardtius hippoglossoides* | 3 | 1 | 0 | 0 |
| *Sebastes mentella* | 0 | 1 | 0 | 1 |
| *Sprattus sprattus* | 1 | 1 | 1 | 1 |
| *Trisopterus esmarkii* | 9 | 12 | 16 | 30 |

Numbers correspond to the percentage of the total study area in which the species dominate the community (projected species' highest biomass to total biomass).

northern Barents Sea, with doubling of species richness around Svalbard and the north coast of Norway. No increases, and even slight decreases in richness were projected in the deepest part of the Barents Sea, at the Bear Island Trench (Fig. 2A). Smaller increases in species richness were projected in the centre of the North Sea and small declines elsewhere in the North Sea and southern Norwegian Sea.

Relative dominance (%) was weakly inversely correlated with species richness (Pearson = −0.09, $p < 0.01$) and accordingly, we projected declines in percentage of relative dominance in the northern and eastern Barents Sea (Fig. 2B). Species relative dominance changed abruptly for polar cod (*Boreogadus saida*), which went from dominating in 29% of the study area under present-day conditions, to a lack of dominance in any grid cell in the study area under the high emission pathway in 2100 (Table 1). The species that increased most notably their relative dominance in accordance were Norway pout (*Trisopterus* esmarkii), blue whiting (*Micromesistius* poutassou), and whiting (*Merlangius merlangus*) (Table 1).

### Individual species geographic range shifts

Species were generally projected to increase their distribution range and biomass with high emission scenarios, particularly in their core range (Fig. 3, Supplementary Data 1). However, this was not a homogeneous response, and differences between contracting and expanding species increased under the high emission scenarios (Figs. 3 and 4 and Supplementary Figs. 4 and 5). The few Arctic species for which biomass models were considered adequate ($n = 3$), and the Arctic-boreal species ($n = 8$) were projected to strongly decline across all scenarios, while boreal ($n = 40$) and warmer-water species ($n = 5$) were projected to expand across the whole study area (Fig. 4). However, present-day projected biomass of currently abundant species (mostly polar cod *Boreogadus saida*, and capelin *Mallotus villosus*) was not compensated by expanding boreal species in future scenarios, leading to an overall decline in biomass with climate warming (Fig. 5).

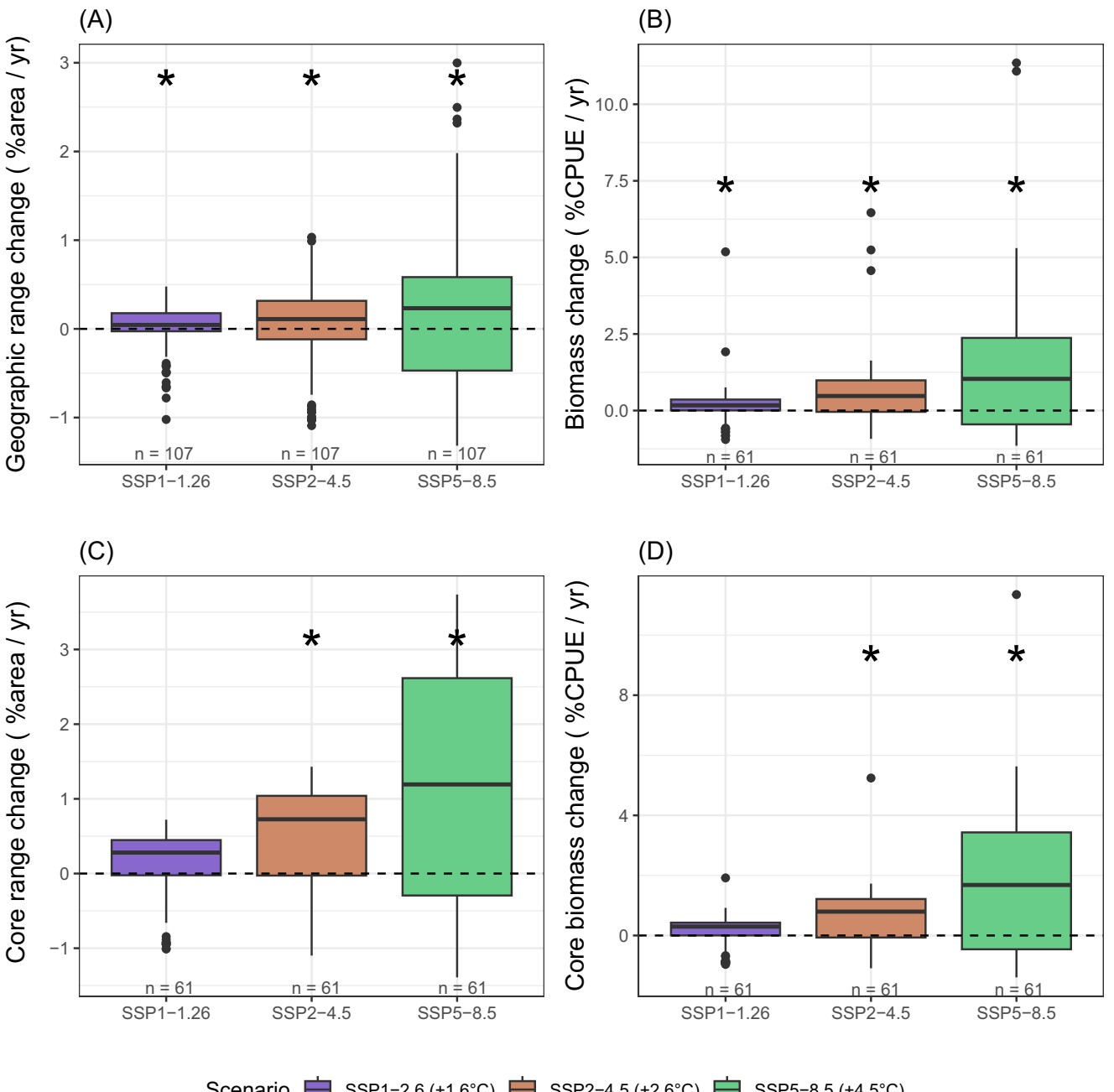

**Fig. 3 | Range and biomass change.** Projected rate of change in **A** species' geographic range change, **B** species' biomass, **C** species' core range, and **D** species' core biomass, at each socioeconomic pathway between 2010 and 2100. Boxplots are a standardized way of displaying the distribution of the data by showing the median percentage change of all species (black line) while box limits correspond to the interquantile range (IQR). Whiskers correspond to maximum and minimum values, calculated as Q3 + 1.5xIQR and Q1 – 1.5xIQR, and points correspond to outliers. Asteriscs indicate signifiant difference from 0 (Two sided Wilcoxon rank test, $p < 0.05$, see Source Data for individual test results). Biomass units are in catch per unit effort (CPUE fish/min).

Among the 7 studied traits, species zoogeography was the best variable in explaining species rate of change of range, core range, and overall biomass. Arctic and boreal-arctic species were projected to decline in the future, and boreal, temperate, subtropical, and deep-water species, to increase (Fig. 6) (multiple linear regression, $p < 0.05$). Apart from species zoogeography, species trophic level showed a positive effect in species' range extent, maximum length showed a positive effect in species' biomass, and maximum depth showed a positive effect in species' core range (multiple linear regression, $p < 0.05$).

Community-wide range shifts northwards and eastwards were projected across species between 2010 and 2100 (Supplementary

Data 2 and 3). Range shifts increased with socioeconomic pathways from a mean of 0.9 km yr$^{-1}$ northwards and 0.3 km yr$^{-1}$ eastwards under SSP1-1.26, to 3.2 km yr$^{-1}$ and 1.1 km yr$^{-1}$ northwards and eastwards, respectively, under SSP5-8.5 (Supplementary Fig. 6A). Smaller shifts were projected for biomass-weighted centroid shifts, from a mean of 1.0 and 0.8 km yr$^{-1}$ northwards and eastwards under SSP1-1.26, respectively, to 3.7 km yr$^{-1}$, and 2.1 km yr$^{-1}$, respectively, under SSP5-8.5 (Supplementary Fig. 6B). The highest shifts were detected in species' core range, containing the top 10% of species biomass, for which projected shifts were 1.1 and 0.5 km yr$^{-1}$ northwards and eastwards, respectively, under SSP1-1.26, to 4.8 km yr$^{-1}$

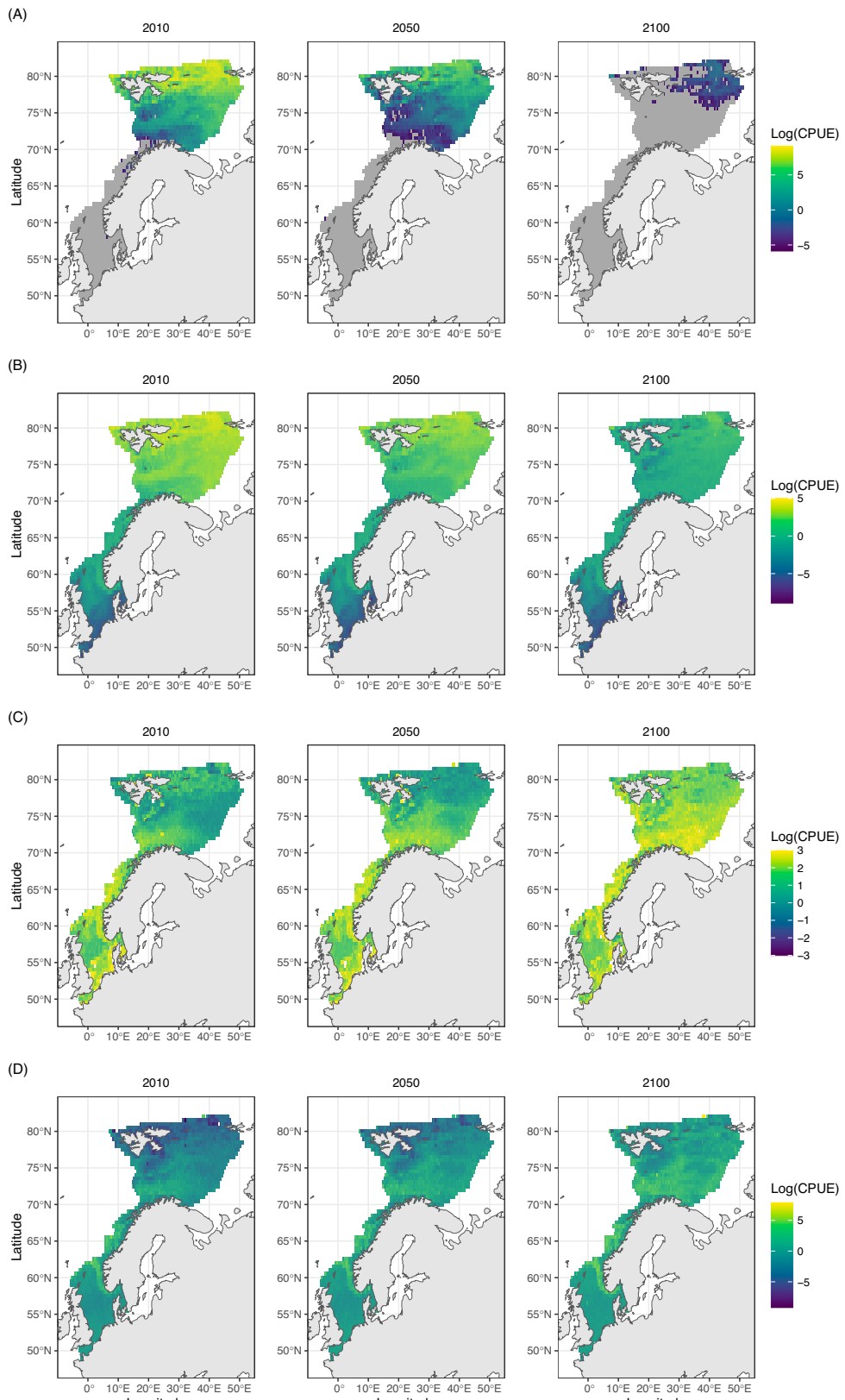

**Fig. 4 | Biomass distribution change per zoogeographic group.** Mean future biomass projections under SSP5-8.5 for (**A**) arctic (*n* = 3 species, grey is study area), **B** arctic–boreal (*n* = 8), **C** boreal (*n* = 40), and **D** temperate and subtropical (*n* = 5) species. Projections of SSP1-2.6 and SSP2-4.5 are show in supplementary Figs. 4 and 5. Log CPUE corresponds to log of catch in the survey per unit of effort (Log fish/min).

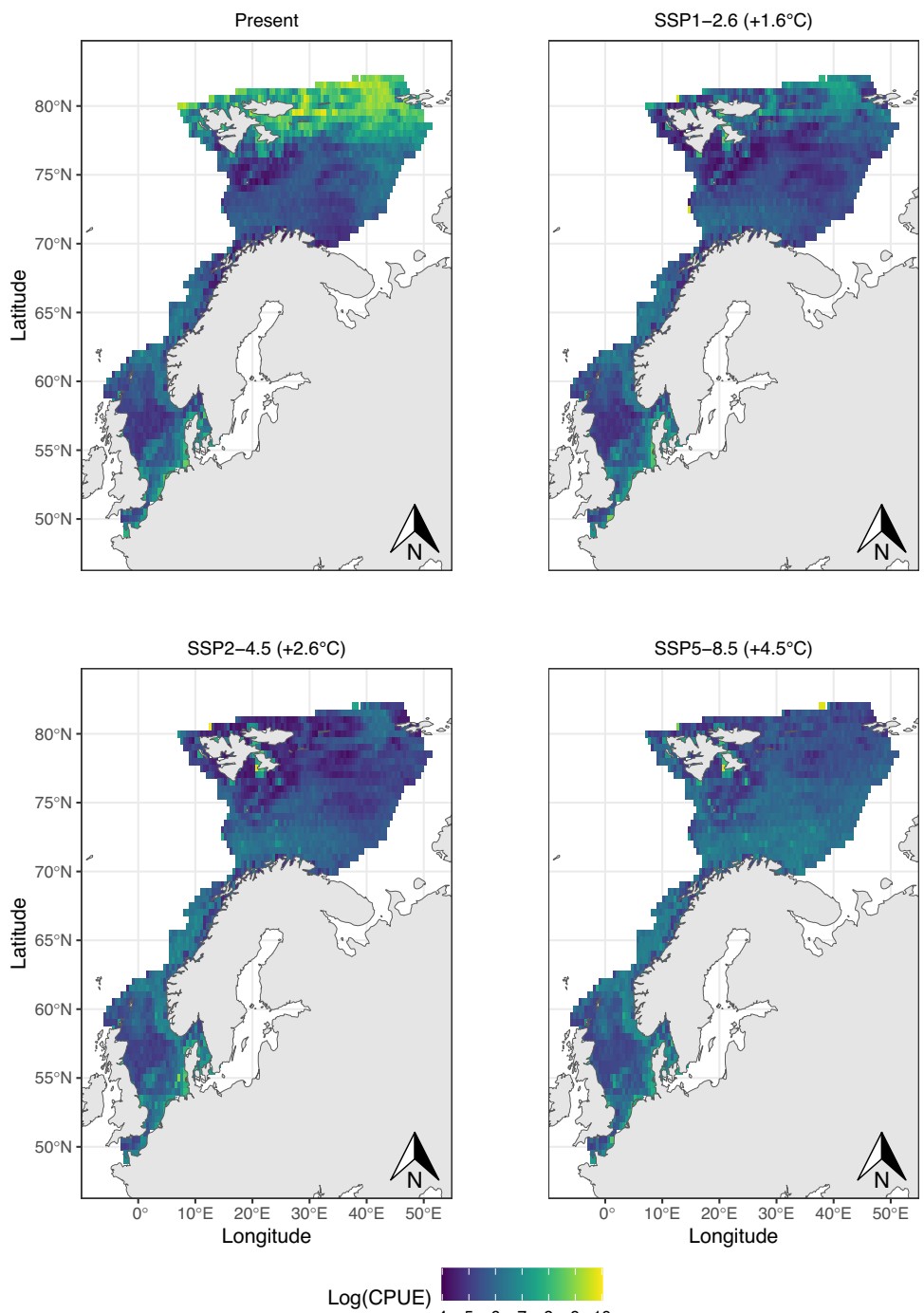

**Fig. 5 | Projected overall biomass in present-day conditions, and in 2100 under future scenarios.** Overall biomass was calculated as the logarithm of the sum of each species biomass in the survey (Log fish/min, $n = 61$).

and 3.2 km yr$^{-1}$, respectively, under SSP5-8.5 (Supplementary Fig. 6C).

### Geographic range fragmentation

Our results do not show clear trends in habitat fragmentation across species. Although we projected increasing number of polygons per species with increasing socioeconomic pathway, and declines in polygon area (Fig. 7, Supplementary Data 4), these trends are driven by (1) Arctic species declining in polygon area, and (2) warmer-water species increasing in number of polygons (Supplementary Fig. 7). The combination of these two parameters in the same species would lead to increased habitat fragmentation, but each of these processes in

different zoogeographic groups suggest no clear trends in habitat fragmentation.

### Discussion

We project a drastic reduction of Arctic and boreal-Arctic marine fishes' ranges and biomasses in the study area with increasingly pessimistic greenhouse gas emission pathways. This may result in the local extirpation of several of those species by within the next decades. Although the expansion of several boreal and warmer-water species leads to an increase in species richness, the present-day projected biomass of Arctic species is not fully replaced by expanding species, resulting in biomass declines across future scenarios. Global ensemble

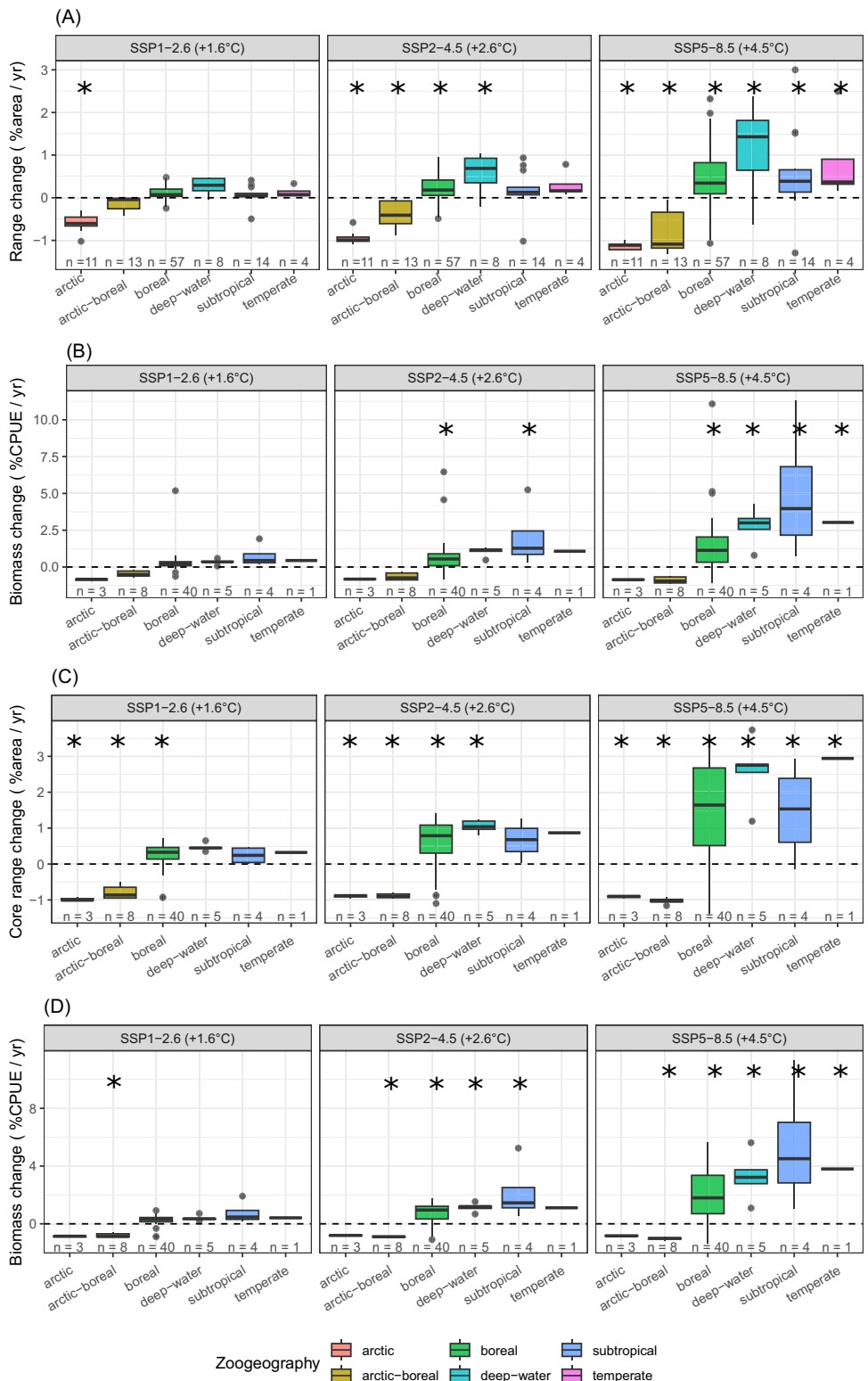

**Fig. 6 | Biomass and range annual change per zoogeographic group.** Projected effect of species zoogeography on annual rate of change from 2010 to 2100 in species' (**A**) geographic range, **B** biomass, **C** core range, and **D** core biomass at each socioeconomic pathway. Boxplot black line shows the median change, while box limits correspond to the interquantile range (IQR). Whiskers correspond to maximum and minimum values, calculated as Q3 + 1.5xIQR and Q1 − 1.5xIQR, and points correspond to outliers. Asterisks represent clear effects of each zoogeographic class at a particular shared socioeconomic pathway (multiple linear regression, $p < 0.05$, see Source Data for individual test results). Biomass units are in catch in the survey per unit effort (CPUE fish/min).

mechanistic modelling efforts conducted in recent years predict increases in consumers biomass in the high Arctic across several taxa, but taxonomic resolution remains a barrier to further interpretation and uncertainty is very high[35]. Projected increases in biomass,

however, could accumulate in different components of the community, and do not necessarily conflict with our projections of overall fish biomass reductions. Moreover, we did not include the effect of fishing impacts in our study, which can act synergistically with climate change

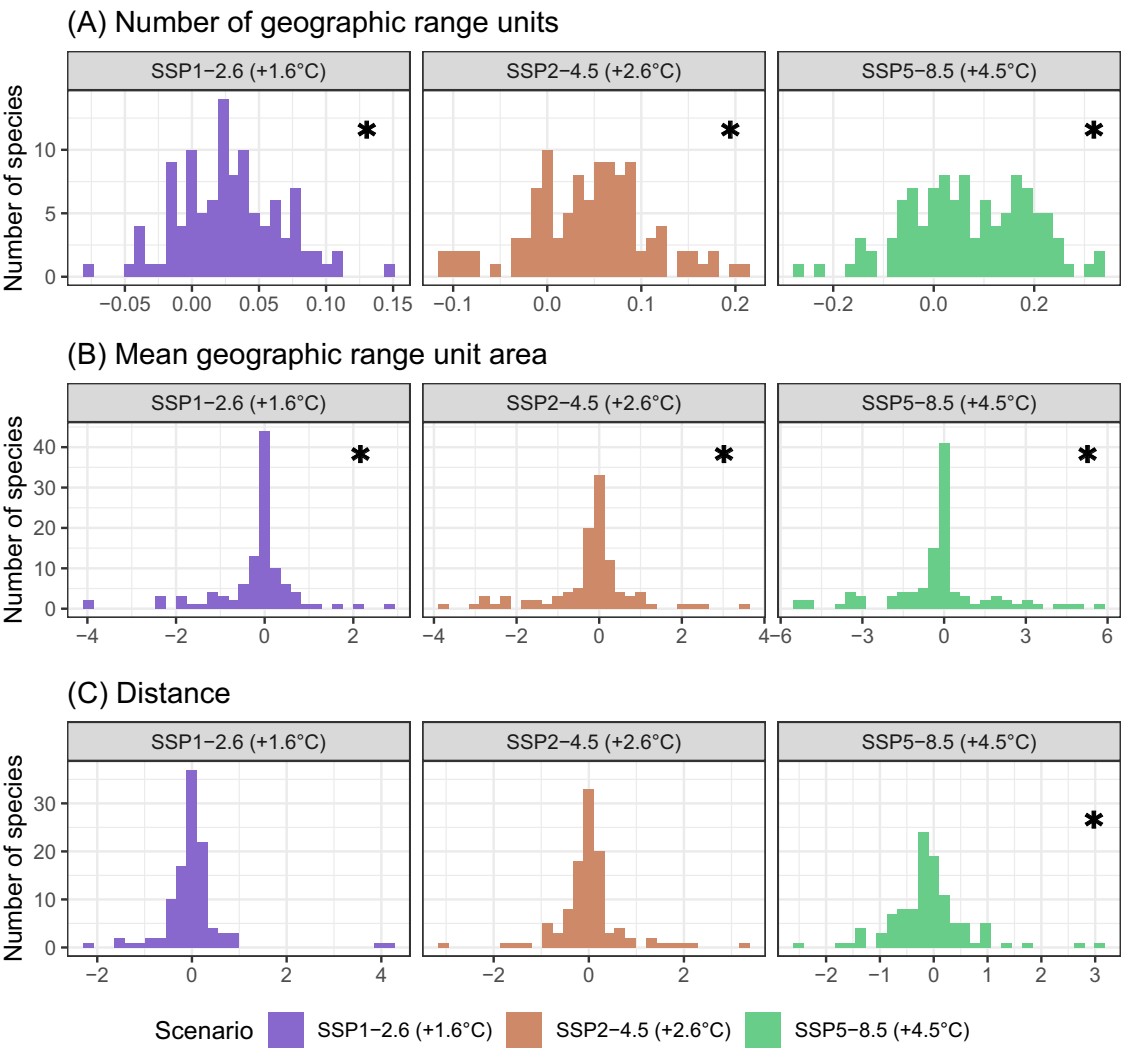

**Fig. 7 | Geographic range fragmentation change per species and shared socioeconomic pathway from 2010 to 2100.** Histograms correspond to species' (**A**) rate of change in mean geographic range unit area. **B** Rate of change in mean area between geographic range units, and **C** rate of change in distance between geographic range units. Geographic range untis are the individual polygons that compose a species' projected range. Asterisks indicate significant median different from 0 (Two sided Wilcoxon rank test, $p < 0.05$, see Source Data for individual test results).

and therefore is an important element that needs to be considered in future research, when future predictions of fleet behaviour are developed[35].

Polar species are at higher risk of extinction than species from lower latitudes[34], and in the case of Arctic demersal fishes, the lack of continental shelf further north than the Barents Sea could represent an additional pressure that limits available habitat at northern latitudes[33,36]. Thus, the local extirpation of Arctic and Boreal-Arctic demersal species from the Barents Sea could place those species at high risk of global extinction. Among species showing decreasing trends, we project a total extirpation of one boreal-arctic species, the Atlantic poacher (*Leptagonus decagonus*), under the high emissions scenario. Two out of three Arctic species, the bigeye sculpin (*Triglops nybelini*) and the pale eelpout (*Lycodes pallidus*), are projected to suffer local extirpations in the study area not only under the high emissions scenario, but also under the intermediate scenario. The third Arctic species with modelled biomass, polar cod (*Boreogadus saida*), is projected to lose its relative dominance in all its occurrence area, to the practical disappearance from the study area by the end of the century under the high emissions scenario (residual biomass projected), though strong declines of over 50% in area and biomass are projected under all scenarios. The pelagic nature of polar cod may

allow adult individuals to shift to other Arctic regions, some of which may become more suitable for the species in the future[37], but its early life stages are highly linked to sea-ice cover, and its recruitment is predicted to collapse with further sea-ice cover reduction[38]. Whether these and other Arctic or Arctic-boreal species will be able to find refugia by moving to other regions where warming is happening at a slightly slower rate remains to be investigated. This high sensitivity of Arctic and boreal-Arctic fish species to climate warming contrasts with the surprising robustness predicted for Arctic benthic taxa[39], which could lead to novel species interactions with overlapping predicted ranges[40]. However, whether higher trophic levels will be able to shift to other food-sources if Arctic keystone species such as polar cod disappear from the Barents Sea is still unknown (i.e., ringed seals (*Phoca hispida*) are highly dependent on polar cod[41,42]).

Overall, our results are in line with the reported ongoing global redistribution of marine species, with increasing richness at higher latitudes[1]. Our projections suggest that these geographic shifts will lead to biomass increases of several boreal and warmer-water species, and declines of Arctic and Boreal-Arctic species without marked changes in species range fragmentation. These shifts will further contribute to the ongoing borealization of the fish community, and the positive richness trend in the area[30,43]. We predict

that this increase in richness will be accompanied by a shift in the identity of the dominant species, indicating a change in the community which is expected but not always the case[44,45], and a general decline in relative dominance. This could have implications in future modelling studies of those communities, as increases in species richness and abundance improve the predictability of community properties[45]. Moreover, the number of dominant species across the study area declined with time, from 10 species in present-day conditions to 6 species under the high emissions scenario, which points towards a homogenization of the community by the end of the century. Although we project slight declines in richness at lower latitudes of the study area, the interpretation of richness and relative dominance changes in southern regions of the study areas require caution, considering that species expanding their range from outside the study area into the study area are to be expected, but are here not accounted for. Overall, we project future communities with a more balanced share of biomass, higher richness, and lower spatial variation in the dominant species if climate warming exceeds 1.5 °C.

Our results also show that current predictability of Arctic species is particularly challenging due to little data available for model calibration for several Arctic species, suggesting that borrowing information from phylogenetically close species is not enough to obtain informative predictions for those species. Furthermore, SDMs often perform substantially better predicting species static patterns, than predicting changes[46]. To properly validate the models for predicting species range shifts, their predictive performance should not be tested on their ability to predict static distributions, but on their ability to predict distributional changes[46]. However, this requires temporally independent data at high-enough spatial resolution to detect distributional changes, and this is missing for most of the species included in this study. Therefore, we believe that there is an urgent need for biomass data collection of shifting species at fine spatial resolution, particularly at species range edges and particularly focusing on Arctic demersal species, which are likely at the front of species risk of global extinction.

The projected increase in biomass of several warmer-water species, linked to the expansion of species from lower latitudes, may represent novel fishing opportunities. For example, ling (*Molva molva*), monkfish (*Lophius budegassa* & *Lophius piscatorius*), whiting (*Merlangius merlangus*) or haddock (*Melanogrammus aeglefinus*) are here projected to increase in range and biomass, while mackerel (*Scomber scomnrus*), a species whose expansion northwards has already led to transboundary conflicts in the northeast Atlantic[47], is projected to increase in range at northern latitudes. Interestingly, our projections suggest that cod (*Gadus morhua*), currently an expanding species in the Barents Sea[16], will slightly decline in biomass by the end of the century. This is in line with similar predictions in the North Sea, where although several species of fisheries interest were projected to expand their suitable habitat, cod showed a reduction in habitat suitability by mid-century[18]. This suggests that for some boreal species, future warming may be of enough magnitude to revert their current expansion trends.

Previous studies in the North Sea, have also identified bottom temperature as the main environmental variable shaping marine fish communities[48,49], as have other studies elsewhere[50–52]. Similarly, we identified depth and bottom temperature as the most relevant predictive variables in our study, although the relevance of phytoplankton and dissolved oxygen could have been hindered by the lower resolution of these two variables, which were obtained from a different Copernicus dataset than the rest. Moreover, studies in the North Sea have widely reported climate-warming induced species northward shifts[8,29], which resulted in local species richness' increases[53], although some species of commercial interest (e.g., Atlantic cod) may decline in the mid term[18,54]. Our projections partly point in this direction and

show regional slight increases in richness in the North Sea, though also some regional declines. However, special caution is required when interpreting projections in the North Sea for three reasons. First, because future climate in the region has no analogue anywhere in the model calibration and future North Sea conditions are not represented in the study area at present-day conditions, this fraction of the environmental space is poorly sampled during the model calibration. Projections for the less sampled parts of the environmental space are considered less reliable and should be interpreted with greater caution[55,56]. Second, the historical run of the global earth system model used to obtain environmental data for future projections shows discrepancies with the fitting of environmental data in the North Sea, as the correlation analysis in the supplementary material shows. Finally, in considering richness and dominance estimates, it is highly likely that species expanding their range from outside the study area into the study area concentrate in the North Sea, which is our lowermost region, and these would not be accounted for here. For these reasons, we advise caution in interpreting our results in the southernmost region of this study.

Facing the challenges posed by climate warming, future fishery management strategies must consider shifts in species biomass dynamics and distributions. However, current day modelling techniques and data collection need to be improved for many species to be able to achieve this crucial objective. For example, better modelling of species biomass in our area could help to prevent fisheries conflicts among several economic zones, due to transboundary stock shifts under climate warming[9,57,58]. Furthermore, fisheries will also affect future marine fish communities, as they have affected marine fish communities in the North and Barents Seas for decades[43,59], and current management decisions will contribute to shape fisheries resources in the future, which future modelling approaches will need to include. As such, embracing adaptive management strategies that account for the evolving dynamics of marine ecosystems and fisheries resources is imperative to ensure the sustainability and resilience of our oceans for generations to come.

## Methods
### Study area and fish biomass data
Fish biomass, as relative abundance (catch per unit effort CPUE) was obtained from bottom trawling data collated within the FishGlob database of fish records and biomass standardised with sampling effort, from several international trawl surveys[60] (Fig. 1). The data was filtered to the study area (North Sea to Barents Sea), and to the period 2004–2022. This included data from three surveys: the NS-IBTS in the North Sea, the Norwegian Sea coastal survey in the Norwegian Sea and the Nor-BTS in the Barents Sea[60]. We restricted the analysis to Campelen and GOV trawls, the two main gears used in the North Sea (GOV) and in the Barents Sea (campelen trawling), all gears were equipped with 20 mm mesh size nets bottom trawls, and each haul catch was standardised by effort[60].

We restricted the data to the continental shelf, by eliminating the few hauls that were sampled deeper than 500 m (<1% of the hauls), and we eliminated very rare species that were sampled in less than 100 hauls, or in less than 10 years. Finally, some trawls (<1%) were eliminated due to lack of environmental data. The final database included 16,345 unique hauls and included 107 fish species (Supplementary Data 5).

### Environmental variables
We selected those physical and biological variables that, based on previous knowledge and expert opinion, are thought to drive the spatial-temporal variability in demersal fish species biomass (Husson et al., 2020). We included (1) surface and (2) bottom water temperature, (3) sea ice concentration for species occurring in areas with sea ice (i.e., not included for strictly temperate species whose model

showed very poor convergence for sea ice parameters because no overlap between occurrence/biomass and sea ice, (Supplementary data 5), (4,5) currents (northward and eastward components), (6) bottom dissolved oxygen concentration, (7) phytoplankton concentration, and (8) water depth.

Sea ice concentration, surface and bottom temperatures, and northward and eastward current components were obtained from the Global Ocean Physics Reanalysis at a resolution of $0.08° \times 0.08°$, while bottom dissolved oxygen, and bottom primary productivity were obtained from the Global Ocean Biogeochemistry Hindcast at a resolution of $0.25° \times 0.25°$, both of which were available through the Marine Copernicus repository[61,62]. Bottom depth was obtained from BioOracle at a resolution of $0.08°$[63]. Environmental information was extracted for each sampling point corresponding to the monthly mean of each survey month, except for sea ice, where the annual mean was preferred, because winter sea ice dynamics can highly influence the populations of several Barents Sea marine fishes throughout the year (Supplementary Table 1).

To remove co-linear variables, which can increase uncertainty and decrease statistical power of the models[64], we calculated the Variance Inflated Factor (VIF), and eliminated all variables using a conservative threshold VIF greater than 4[65]. This led us to eliminate surface temperature, as it was highly correlated with bottom temperature, leaving 7 environmental variables to include in the model. Moreover, for bottom temperature and depth, we included their second-order polynomial responses, to allow a bell-shape response of species distribution and biomass to these variables[66].

## Future environmental layers
Future mean annual environmental layers were obtained from the second version of the Institute Pierre Simon Laplace climate model (IPSL- CMIP6)[67], which is the only model within the Coupled Model Intercomparison Project Phase 6 (CMIP6) that contained all our predictive variables. We used future environmental data from three different shared socioeconomic pathways (SSPs): the most optimistic 'high mitigation' scenario reflecting sustainable development and social justice, where the probability of exceeding +2 °C by 2100 is kept below 33% (SSP1-2.6, + 1.6 °C by 2100), an 'intermediate scenario' (SSP2-4.5, + 2.6 °C by 2100), and a scenario reflecting a world of rapid growth and without restrictions on economic production and the use of energy, the 'high emissions' scenario (SSP5-8.5, + 4.5 °C by 2100)[68].

Mean annual environmental data layers for each scenario (SSP1-2.6, SSP2-4.5 and SSP5-8.5) were extracted in 10-year increments and used for predicting species' distributions. For comparison with the present-day conditions, we used the mean between 2010:2013 to represent present-day conditions, leading to an overall of 25 time periods (8 future periods × 3 scenarios, and 1 present-day scenario).

## Statistical modelling
A two-part hurdle model was used to model the distribution of demersal fish biomass dealing with the excess number zeroes in the dataset[69]. In this procedure, a binomial model with a probit distribution was used to project the probability of species' occurrence, and a separate model with a Gaussian distribution was fit to species' log transformed catch per unit effort, to project species' biomass. This was done using the *Hierarchical Modelling of Species Communities* JSDM approach (in the 'Hmsc' package in R, Tikhonov et al.[70]).

In both models (binomial and gaussian), spatial autocorrelation was accounted for by including a spatially explicit random effect using Gaussian Predictive Processes (GPP assuming that information on the spatial structure of the data can be summarized at a smaller number of (in our case 183) 'knot' locations[71]. The inclusion of spatial autocorrelation when fitting the model is highly recommended to improve the estimation of the model coefficients[72].

Although we excluded the rarest species from the analysis, most of the species remaining can still be considered rare (65% were present in <10% of the hauls). This poses a challenge in estimating the realized niche of those species, which only have few points for estimating the limits of their environmental niche. However, species inhabit environments that share some similarities with those of their close relatives, because they follow niche conservatism to some degree[73]. In the HMSC framework, the statistical relationships between species occurrences or biomass with the environment are integrated through a hierarchical structure that allowed us to determine to what extent environmental filtering is structured by species phylogenetic relationship, and species traits[70]. This is done by including a phylogenetic correlation parameter *rho* that measures if the residual variation of species responses to the environment is phylogenetically structured. For this reason, we built a basic taxonomic tree using the NCBI Common Tree software, available through its website[74], which is the best available proxy for phylogenetic relatedness when species phylogenetic data are not available with very similar relationships to formal taxonomic classifications. Strong phylogenetic signals may point to response traits that have not been specifically accounted for in the model.

We analysed projected species distributions to study whether some species sharing traits responded similarly in range expansion and/or contraction using backward selection multiple regression analysis. We selected eight species' biological traits that could be related to species expansion potential[29,75] including five functional traits: (1) maximum length (cm), (2) age at maturity (years), (3) fecundity (number of eggs), (4) habitat (demersal or pelagic), and (5) trophic level; one physiological trait (6) preferred temperature (°C); and one bathymetric trait (7) maximum depth. Traits for each demersal fish species were obtained from FishBase[76]. We finally created a zoogeography trait (8) assigning a general climatic classification for each species, of the following categories: 'Arctic', 'Arctic-Boreal', 'Boreal', or 'Deepwater', as classified in Mecklenburg et al.[77], or from FishBase when the species were not present in the former reference (i.e., not present in Arctic latitudes), adding the categories 'subtropical' and 'temperate' to the list of possible categories[76].

Model fitting was conducted using the Markov Chain Monte Carlo (MCMC) implemented in Hmsc. Model convergence was assessed using the Gelman-Rubin Potential Scale Reduction Factor[78]. Four MCMC chains were run, each collecting 250 samples, applying a thinning of 500, and the first 62 500 runs were discarded as burn in. The MCMC convergence was satisfactory, as indicated by a mean scale reduction factor of all parameters <1.1, and the effective sample size of the MCMC was close to the number of posterior samples, indicating no major issues of sample autocorrelation. Model goodness of fit was then assessed by computing the overall explanatory capability (calculated from the species' data used in the HMSC model fitting) as the mean AUC and $r^2$ value across all species-specific values. To evaluate the predictive performance of the model, a five-fold cross-validation was undertaken (i.e., assessing the HMSC model fit using the withheld data from each fold). The model fitting, calibration and validation was done using the 'Hmsc' package in R[70], and code from[66,79].

The explanatory power of the model had a mean AUC of 0.97 for the presence-absence model, and an $R^2$ of 0.54 for the biomass model, while the mean predictive power from a five-fold cross validation was lower (AUC of 0.88, and a mean $R^2$ of 0.12) (Supplementary data 5). Species richness projected using present-day layers was significantly correlated with surveyed species richness (Pearson r = 0.4, p < 0.01).

To show the discrepancy between the fitting and the predictive datasets (fitting was done with Copernicus datasets, and projections with IPSL global earth model), we conducted a correlation analysis between both sources of SBT monthly averages, in the overlapping period of the IPSL historical run, and the SBT data from Copernicus for the coordinates included in this study. This includes all our data between 2004 and 2014. We show that both datasets are significantly

correlated, but present discrepancies in the North Sea (Person correlation = 0.61 Pearson correlation excluding North Sea = 0.85) (Supplementary Fig. 8). Moreover, we conducted a multivariate environmental similarity surface (MESS) analysis using the MESS() function from the modEvA package in R[80], to assess whether the 'environmental space' in our projections was accordingly sampled during the model training (Supplementary Fig. 9). The 'environmental space' is the multidimensional space produced by considering each of the environmental variables as a dimension. Projections in poorly sampled parts of the environmental space are considered less reliable (strongly negative MESS values), and should be interpreted with greater caution[56,81].

Finally, we examined the patterns of species co-occurrences at the level of the spatial random effect. The co-occurrence of specific species is drawn from the covariance structure of the model residuals once the fixed environmental effects have been considered. This analysis reveals pairs of species that either co-occur more frequently or less frequently than expected by random chance, which can partly represent biotic interactions[25,82].

Biomass models: From the initial 107 species included in the model, we restricted all biomass-based analyses to species with > 0.05 mean $R^2$ in the five-fold cross validation biomass model.

Although explaining only 5% of the variance of the data may seem a rather poor fit, two things need to be considered: First, the distribution of the biomass is determined by the occurrence model which shows substantially better predictive performance. Second, the threshold of 5% is arbitrary, but not random. All models with a CV $R^2$ higher than 0% are informative, but because every fold of the CV provides a different value, and most of the average values close to 0% have some folds with negative percentages, we chose a more conservative threshold of 5%. This resulted in the inclusion of 61 species with mean $R^2$ of 0.22, ranging from 0.05 to 0.53 (Supplementary data 5). We multiplied the projections from the gaussian log biomass model with the projections from the binomial model to obtain the final biomass projections. Then, we assigned 0 to all projected biomass lower than the minimum value recorded in the dataset, divided by 2. This was done to assign a threshold for presence-absence of species that is biologically meaningful (minimum recorded) while assuming certain presence below detectability (divided by 2).

Occurrence models: All 107 species included in our study had reasonable predictive performance of presence-absence models (lowest AUC = 0.67 in five-fold cross validation). For this reason, we included all species in those analysis that required only presence-absence information (i.e., changes in species geographic range, and species richness). To threshold presence-absence in those analyses, we calculated the threshold of probability of occurrence that maximised the True Skill Statistic per species[83], and we used that species-specific threshold to assign presence or absence of each species across the study area.

Spatial projections: We projected geographic distributions of species' occurrence and biomass across the period 2010:2013 (historic reference, which we refer to as 'present-day' projections) and from 2030 to 2100 every 10th year for three different possible climate change scenarios (SSP1-2.6, SSP2-4.5 and SSP5-8.5). Although our model included spatial processes for estimating its parameters (using the GPP methodology explained above), future projections excluded the spatial random effect. This was done consciously, to avoid extrapolating estimated spatial structures that may not persist under future climatic conditions.

### Geographic range metrics
Four species geographic distribution indicators were used to explore changes in distribution over time for each climate change scenario. First, the 'range' of a species was measured as the area (km²) of projected occurrence (based on the thresholded spatial projections from the presence-absence model). The second measure 'biomass' quantified the total projected species' log CPUE across the study area (based on the thresholded spatial projections from the hurdle model). Third, the 'core range' represented areas with the highest biomass. Core range was identified using the 90th quantile of present-day species' biomass (that is we selected cells that contain the top 10% of CPUE). Future core range was estimated by selecting all areas where future CPUE was ≥ than the present-day 90th quantile CPUE value. Finally, the fourth measure 'core biomass' refers to the total biomass (CPUE) within the core range. These measures were calculated by projecting the model into the equal-area Lambert azimuthal projection, and we studied their rate of change regressing them with time (using linear regression). The analysis of the data was conducted using the 'raster' and 'rgdal' packages in R, and plotting was done using tidyverse and ggspatial, while the MESS analysis was conducted using the modEvA package[80,84–88].

To study potential changes in geographic range fragmentation, we converted each species' projected range to polygons, and calculated the number of polygons, the mean area, and the mean distance between polygons of each species (Supplementary Data 4). Many small polygons with large mean distance between them would represent a very fragmented range, while few, big polygons, close to each other, would represent a less fragmented range.

### Species richness and relative dominance
Species richness was calculated by summing all projected probabilities of occurrences across species ($n = 107$)[89]. Dominance was calculated as the percent contribution of the highest species' CPUE to the total CPUE (sum of all CPUE of all species) at each cell, and dominant species are the species with the highest biomass in each cell.

### Reporting summary
Further information on research design is available in the Nature Portfolio Reporting Summary linked to this article.

## Data availability
The data used in this study was obtained from bottom trawling data collated within the FishGlob[60] (Accessible at https://github.com/AquaAuma/fishglob_data). The Norwegian Sea section of this data is no longer available in FishGlob, and needs to be directly asked for to the Norwegian Marine Data Centre (https://metadata.nmdc.no/metadata-api/landingpage/f77112db062b5924d079a54b311260fb). The environmental data used for calibrating the model came from the 'Global Ocean Physics Reanalysis' and the 'Global Ocean Biogeochemistry Hindcast' both of which were available through the Marine Copernicus repository[61,62] at https://data.marine.copernicus.eu/products. Bottom depth was obtained from BioOracle[63] at https://www.bio-oracle.org. The environmental data used for future projections came from the second version of the IPSL climate model (IPSL-CMIP6)[67], and is fully accessible as well at https://esgf-data.dkrz.de/search/cmip6-dkrz/. World administrative boundaries polygons are available from opendatasoft, accessible at: https://public.opendatasoft.com/explore/dataset/world-administrative-boundaries/information[90]. The data generated in this study, and used for Figs. 3, 6 and 7 is provided in the Source Data file, while the output of all regression analysis are available in the Supplementary Data files, as well as the individual species range and biomass projections. The trait database gathered is available at https://github.com/CescGV/JSDM-Barents-Norwegian-North[91]. Source data are provided with this paper.

## Code availability
Code used for this publication is available at https://github.com/CescGV/JSDM-Barents-Norwegian-North[91].

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

## Acknowledgements
C.G.V. would like to thank Lars Martin Jakt from Nord University, for introducing me to bash, tmux and emacs. Thanks to Aleksandra Elbakian for always being supportive and providing us with great input. MC would like to acknowledge partial funding from the Spanish National Project ProOceans (PID2020–118097RB-I00) and institutional support of the 'Severo Ochoa Centre of Excellence' accreditation (CEX2019-000928-S).

## Author contributions
C.G.V.: Conceptualization, Visualisation, Data clenaing, model construction and analysis, writing – original draft; M.J.C.: Conceptualization, writing – review & editing; M.C.: Conceptualization, Writing – review & editing; A.J.: Conceptualization, Writing – review & editing; H.R.: Conceptualization, Writing – review & editing; F.S.: Conceptualization, model construction, Writing – review & editing.

## Funding

## Competing interests
The authors declare no competing interests.
