## [Peer Review File · Nature Communications]

Future trends of marine fish biomass distributions from the North Sea to the Barents SeaREVIEWER COMMENTS

Reviewer #1 (Remarks to the Author):

With great interest I reviewed your manuscript "Future trends of marine fish biomass distributions from the North Sea to the Barents Sea: Arctic species at stake" for Nature Communications. Based on a large number of scientific trawls, state-of-the-art joint species distribution models, and climate change projections, the authors offer insight in expected changes in the biomass, richness and distribution of demersal fish in the North Sea and further northwards.

Since the primary aim for the reviewer is to judge the soundness of the research, I focus on the significance of this manuscript along the proposed reviewer guidelines.

Key results

This study uses unmatched data volumes and novel species distribution models, which include traits and taxonomic relatedness, to project future fish distributions across an unrivalled scale for three different climate change scenarios. Such work is of critical importance to inform management and conservation efforts, since it shows that currently defined MPAs might not be relevant in future years and that some species, such as Arctic Cod, have no habitat left to escape to and are likely to go nearly extinct. Moreover, predicted shifts in richness and biomass are shown, although these do not compensate for predicted loss of Arctic species.

Significance

The topic of shifting distributions is not new to science and has been explored for North Sea and Arctic marine areas separately repeatedly. But, this is the first work that uses JSDMS to assess shifting demersal fish distributions under future climate scenarios at such a large scale and for so many species. It provides insight in predicted future species distributions that are not available elsewhere. The use of JSDMs is likely to set the standard for future research on climate change and biodiversity patterns.

Validity

In my view, all interpretations and conclusions are supported by the analyses.

Data and methodology

The data and selected methods are at the frontier of what is currently possible with species distribution modelling, since it combines a large amount of data, many species, spatial autocorrelation, traits, and phylogeny in a single model. Data are all available in online repositories, and Rcode to perform the modelling is provided via Github, but also the excellent book by Ovaskainen and colleagues offers detailed information and Rcode to conduct the modelling.

After reading the Methods part, I wondered how the authors dealt with the different gear-types that occur in the FishGlob database. That all gears have a similar mesh size is important, but also the size of the used gear is key, since that influences the catchability of species. In the Manuscript, I could not discover how you solved this, but in your Github Rcode I found an answer (Data prep/S0.1 Data cleaning, L20-23). Please add a sentence or 2 in the Methods (L96-109) on that, since anybody familiar with scientific trawling would question a paper that ignores differences between gears.

Another question emerged after looking at Table 2. How did you define the list of dominant species? I fully understand that you only want to highlight a few, but what criteria is used?

Analytical approach

In the fast-moving field of species distribution modelling, JSDMS (here under the HSMC framework) are a recent development and they represent the current state-of-art and are a leap forward from work presented in papers with a similar goal. A common approach would be to fit models for individual species, and stack these based on some threshold criterion to identify diversity hotspots or distributions. Contrary to JSDMs, this generally ignores traits, species co-occurrences, or phylogeny.

Suggested improvements

- In the Abstract (L20), the time-period is stated as 2004-2022, whereas in the Methods (L100-101) this is 2004-2017.

- L117, this should refer to Table 1
- Statistical modelling. Please add a reference to your Github here
- In the Results (L277-279) you refer to single species depicted in Extended Fig. 1, but these names are not visible there. A solution might be to give each species a number in Table 1, and put these numbers on the x-axis, or not refer to the figure and just present it as a statement.
- L423-426: I guess this sentence needs fixing
- Fig. 2. I guess the legend for the future scenarios of part A is also richness (S)?
- L751: Caption reads 6, should be 7.
- I found the discussion overly focussed on the Arctic and the concluding paragraph (L429-436) is somewhat detached from the rest. I understand what you are trying to tell us (Future management needs to include shifting biomass distributions), but that needs a better presentation. My suggestion would be to include a North Sea focussed paragraph after L428, followed by the rewritten last paragraph.
- Just an idea, that is probably outside the scope of this manuscript, but to give this manuscript more ecological depth, you could present co-occurrence analyses for all scenarios (see Ovaskainen et al. 2017. Ecology Letters 20:561-576 or the HMSC-book) for all species or just for Arctic seas. It might give some insight in changing co-occurrences (as described in Introduction, L47). I have not yet seen such forecasts with JSDMs, and it might show interesting changing patterns if southerly species arrive in the Arctic.

Clarity and context

The manuscript is well written, concise, with context embedded in results from previous work

References

The authors refer previous literature appropriately.

Reviewer #1 (Remarks on code availability):

The code supporting the analysis is available from the first author's Github repository. This code is annotated, supporting its use by others. I suggested to add a reference to this repository in the main document, allowing easy access.

I did not run it since JSDMs usually take a long-time to converge. And the data are available from online repositories as cited, meaning that I would have to download these from the provided repositories, etc.

Reviewer #2 (Remarks to the Author):

This paper presents a compelling investigation into joint species distribution modeling encompassing over 100 fish species within the North Sea/Barents Sea region, projecting outcomes until the end of the century under three distinct emission scenarios. I find the main approach and much of the methodology rigorous and commendable, and the clarity of presentation of the manuscript is noteworthy. However, there are certain methodological concerns that need to be addressed before I can accept the study's future projections and fully endorse the article for publication.

Main issues:

- One of the major issues I have with this research has to do with the different sources of environmental data for model training and future projections. From what I understand these data came from 3 different models, two of them with projections for a past period that were used to train the model (the Copernicus data), and a different model that was used for the future projections under the SSP scenarios. The few times I have attempted to do something like this I have come across compatibility problems; basically I found the data from different models was often not comparable. If I plotted timeseries of my variables over a particular region I often encountered clear discontinuities in the time periods covered by the different data sources. Even using anomalies for the future projections (i.e., adding the difference between the present-day conditions and the future as estimated by a single model to the present-day projections provided by a different model) sometimes still resulted in significant discontinuities, and it was simply not appropriate to combine the different data sources. I would like to see a similar analysis here to convince me that the model data that you are using is compatible. If this is not the case, you may have to limit yourself to using data from the IPSL model only, both for model training and the future projections (also see the next point).
- I have been trying to find out exactly which Copernicus datasets were used for model training. The reference provided in the article only refers to the

“GLOBAL_MULTIYEAR_PHY_001_030” dataset for the physical data, not to the biogeochemical data. However from the description, I believe the second dataset used is the “GLOBAL_MULTIYEAR_BGC_001_029”. If that is the case I’m concerned that they don’t correspond to the same simulation. The biogeochemical model would have needed to rely on data from a physical model, and ideally you’d want to use compatible physics and biogeochemistry data that was generated together. From looking at the documentation of both datasets this does not appear to be the case: the data is available at different resolutions, over a different time period, and have been generated using different versions of the nemo model (version 3.1 and 3.6 respectively). I’m not fully familiar with Copernicus global datasets, so I’m unsure if there is a more suitable alternative or if compatible data just hasn’t been made available yet. Regardless, references for the two individual datasets need to be provided in the text, and if the data is not fully compatible this should be noted. Again, this is not the case for the IPSL model data you used for the future, which would be fully compatible, so the way your model makes use of the environmental data might just not hold with the IPSL dataset, and would make me unsure of your future projections. Because of this I would like to see proof that variable correlation was similar for the IPSL dataset and the training dataset.

- R² is used to test model performance of abundance models, and a cutoff of 0.05 is applied as “good enough” model fit. However explaining only 5% of the variance of the data would seem to me a rather poor fit; is this 0.05 cutoff typically used for abundance SDMs? If so can you provide references in the text? As it is I remain unconvinced that your abundance results are significant.

- I am concerned that the model may be being used to extrapolate far beyond training conditions by considering climate change until the end of the century. This could perhaps be assessed with multivariate environmental similarity surface (MESS) maps to assess where/when the model is predicting in and out of range. Projections would perhaps need to be limited in time or space to stay within training range. Or if extrapolation is indeed needed, model performance would need to be assessed in a different way to better measure the model’s ability to predict to novel conditions (for example via block cross-validation rather than random separation of the data)

Some minor comments:

- Models were trained with monthly data (for most variables), yet they are used to carry out

projections with annual average data. I don't think this is appropriate, as I would expect different correlations to be observed over different scales, both spatial and temporal. In other words, if you trained your models with annual average data instead I would expect the correlations would be different and therefore you'd get different models. If the models are trained with monthly data they need to be used to project with monthly data. If an annual average is preferred, then it should be obtained from the average of the 12 monthly projections, though I think it would be simpler to replace the current annual projections with projections for a particular month. Or alternatively the models could be trained with annual data instead, although this may lower model performance.

- The abstract should specify the period for which projections are made, and the fact that different warming scenarios were considered.

- At the risk of coming across as pedantic, I would suggest that for this work the terms "project" and "projections" should be preferred over "predict" and "predictions", which are frequently used in the document. Admittedly many scientists use these interchangeably, but generally in the climate change community there is a distinction made between the two: projections is appropriate when making inferences for the future that are subject to conditions, as in "if A were to take place, then B is likely to happen" whereas predictions are not subject to future conditions but just to the current status of a system, as if when predicting tomorrow's weather based on today's conditions. For example, long term future climate data are projections, not predictions, as they consider different "scenarios" for the way society and technology may evolve in the future. But as mentioned many scientists consider these terms equivalent.

- It would be nice to see the correlation between the species that your model provides, either in the main text or in supplementary material.

Answer to reviewers:

Future trends of marine fish biomass distributions from the North Sea to the Barents Sea: Arctic species at stake

Cesc Gordó-Vilaseca, Mark John Costello, Marta Coll, Alexander Jüterbock, Henning Reiss, Fabrice Stephenson

First, we would like to thank both reviewers for the positive endorsement of their revisions, and their constructive criticism. We have broadly addressed the comments by providing clarifications and/or new analyses to justify our approach where needed (correlation analysis, MESS analysis, species co-occurrences analysis). Changes have also been made to the wording of the text following suggestions by both reviewers. All changes are provided as track changes in the resubmitted manuscript, all line numbers in the response to reviewers refer to the resubmitted manuscript (with track changes), although we also pasted each section in each answer, for easier review. We answer each of their comments point by point.

Reviewer 1:

With great interest I reviewed your manuscript "Future trends of marine fish biomass distributions from the North Sea to the Barents Sea: Arctic species at stake" for Nature Communications. Based on a large number of scientific trawls, state-of-the-art joint species distribution models, and climate change projections, the authors offer insight in expected changes in the biomass, richness and distribution of demersal fish in the North Sea and further northwards. Since the primary aim for the reviewer is to judge the soundness of the research, I focus on the significance of this manuscript along the proposed reviewer guidelines.

Key results

This study uses unmatched data volumes and novel species distribution models, which include traits and taxonomic relatedness, to project future fish distributions across an unrivalled scale for three different climate change scenarios. Such work is of critical importance to inform management and conservation efforts, since it shows that currently defined MPAs might not be relevant in future years and that some species, such as Arctic Cod, have no habitat left to escape to and are likely to go nearly extinct. Moreover, predicted shifts in richness and biomass are shown, although these do not compensate for predicted loss of Arctic species.

Thank you, it is very reassuring to read this, we agree that it is of critical importance.

Significance

The topic of shifting distributions is not new to science and has been explored for North Sea and Arctic marine areas separately repeatedly. But, this is the first work that uses JSDMS to assess shifting demersal fish distributions under future climate scenarios at such a large scale and for so many species. It provides insight in predicted future species distributions that are not available elsewhere. The use of JSDMs is likely to set the standard for future research on climate change and biodiversity patterns.

Thank you!

Validity

In my view, all interpretations and conclusions are supported by the analyses.

Data and methodology

The data and selected methods are at the frontier of what is currently possible with species distribution modelling, since it combines a large amount of data, many species, spatial autocorrelation, traits, and phylogeny in a single model. Data are all available in online repositories, and Rcode to perform the modelling is provided via Github, but also the excellent book by Ovaskainen and colleagues offers detailed information and Rcode to conduct the modelling.

Thank you!

After reading the Methods part, I wondered how the authors dealt with the different gear-types that occur in the FishGlob database. That all gears have a similar mesh size is important, but also the size of the used gear is key, since that influences the catchability of species. In the Manuscript, I could not discover how you solved this, but in your Github Rcode I found an answer (Data prep/S0.1 Data cleaning, L20-23). Please add a sentence or 2 in the Methods (L96-109) on that, since anybody familiar with scientific trawling would question a paper that ignores differences between gears.

The standardization of the sampling gear is a key point for best fitting of the abundance model. For this reason, and despite FishGlob standardization by effort in all gears, we restricted our analysis to GOV and campelen trawling, the two main gears used in the North Sea (GOV) and in the Barents Sea (campelen trawling). This is now clarified in lines 103-104:

“We restricted the analysis to Campelen and GOV trawls, the two main gears used in the North Sea (GOV) and in the Barents Sea (campelen trawling), all gears equipped with 20 mm mesh size nets bottom trawls, and each haul catch was standardised by effort (Maureaud et al., 2023).”

Another question emerged after looking at Table 2. How did you define the list of dominant species? I fully understand that you only want to highlight a few, but what criteria is used?

Thank you for the question. By dominant, we mean the species with the highest biomass at each cell (lines 304-306) – all species in Table 2 are considered dominant. All the species with the highest biomass in a cell are in that table, and the percentage of area where they dominate is what is shown in the table (summing each column =100% of the area). We have clarified this now in lines 304-306 and 880.

304-306: “Dominance was calculated as the percent contribution of the highest species’ CPUE to the total CPUE (sum of all CPUE of all species) at each cell, and dominant species correspond to the species with the highest biomass at each cell.”

880: “Share of area of each of the dominant species (species with highest biomass at each cell).”

Analytical approach

In the fast-moving field of species distribution modelling, JSDMS (here under the HSMC framework) are a recent development, and they represent the current state-of-art and are a leap forward from work presented in papers with a similar goal. A common approach would be to fit models for individual species, and stack these based on some threshold criterion to identify diversity hotspots or distributions. Contrary to JSDMs, this generally ignores traits, species co-occurrences, or phylogeny.

Thank you, we agree.

Suggested improvements

- In the Abstract (L20), the time-period is stated as 2004-2022, whereas in the Methods (L100-101) this is 2004-2017.

Thank you, this was a mistake. We have changed this now to the correct period throughout (2004-2022).

- L117, this should refer to Table 1

The reviewer is right, this has now been corrected.

- Statistical modelling. Please add a reference to your Github here

This is now added in lines 242-243: "Code used for this publication is available at <https://github.com/CescGV/JSDM-Barents-Norwegian-North>."

- In the Results (L277-279) you refer to single species depicted in Extended Fig. 1, but these names are not visible there. A solution might be to give each species a number in Table 1, and put these numbers on the x-axis, or not refer to the figure and just present it as a statement.

Thank you, we have followed the reviewer's advice and left it as a statement (line 313-318). We refer to the extended figure in the previous sentence (line 315), to show that % explained deviance varied between species. In the new MS, lines 313-318:

"Each of the other variables explained on average less than 1% of the total deviance in both models, although this varied by species (**SM1 Extended figure 1**). For example, phytoplankton concentration explained 20% of variance in the thorny skate (*Amblyraja radiata*) probability of occurrence, and sea ice concentration explained 15% of variance in herring (*Clupea harengus*) CPUE distribution."

- L423-426: I guess this sentence needs fixing

Thank you, this sentence has now been corrected and reads (lines 469-472):

"This is in line with similar predictions in the North Sea, where although several species of fisheries interest were projected to expand their suitable habitat, cod showed a reduction in habitat suitability by mid-century (Townhill et al., 2023)."

- Fig. 2. I guess the legend for the future scenarios of part A is also richness (S)?

Thank you for spotting this, yes, this is now corrected showing dominance and richness.

- L751: Caption reads 6, should be 7.

Thank you, this is now corrected.

- I found the discussion overly focused on the Arctic and the concluding paragraph (L429-436) is somewhat detached from the rest. I understand what you are trying to tell us (Future management needs to include shifting biomass distributions), but that needs a better presentation. My suggestion would be to include a North Sea focused paragraph after L428, followed by the rewritten last paragraph.

It is true that we primarily limited our discussion to Arctic and Boreal latitudes because the observed trends in our study were most novel but also most statistically certain. However, we have followed the reviewer's suggestion and we have now added an additional paragraph explaining the main findings in the North Sea, how these fit within the international literature, and the reasons for which we advise some

caution when interpreting our results in the North Sea. Please see lines 474-511. We have also rewritten the last paragraph to make it clearer (lines 499-511).

“Previous studies in the North Sea, have also identified bottom temperature as the main environmental variable shaping marine fish communities (Montanyès et al., 2023), as have other studies elsewhere (Maynou et al., 2014; Pecuchet et al., 2022; Punzón et al., 2021). Similarly, we identified depth and bottom temperature as the most relevant predictive variables in our study, although the relevance of phytoplankton and dissolved oxygen could have been hindered by the lower resolution of these two variables, which were obtained from a different Copernicus dataset than the rest. Moreover, studies in the North Sea have widely reported climate-warming induced species northward shifts (Gordó-Vilaseca, Pecuchet, et al., 2023; Perry et al., 2005), which resulted in local species richness’ increases (Jones et al., 2023), although some species of commercial interest (i.e., Atlantic cod) may decline in the midterm (Chaikin et al., 2024; Townhill et al., 2023). Our projections partly corroborate these trends; we predict some regional increases in richness in the North Sea, though also some declines. However, special caution is required when interpreting projections in the North Sea for three reasons: firstly, because future climate in the region has no analogue anywhere in the model calibration (that is, future North Sea conditions are not represented in the study area at present-day conditions). Projections for the less sampled parts of the environmental space are considered less reliable and should be interpreted with greater caution (Owens et al., 2013; Stephenson et al., 2020). Secondly, the historical run of the global earth system model used to obtain environmental data for future projections show discrepancies with the environmental data used to fit the models in the North Sea (see a correlation analysis in the supplementary material, Extended Fig 6). Finally, in considering richness and dominance estimates, it is highly likely that species expanding their range from outside the study area into the study area concentrate in the North Sea, which is our lowermost region, and these are not accounted for here.

Future management strategies must consider shifts in species biomass dynamics and distributions when considering the challenges posed by climate change. However, current day modelling techniques and data collection need to be improved for many species to be able to achieve this crucial objective. For now, our projections could inform fisheries management decisions, and could help to prevent fisheries conflicts among several economic zones, due to transboundary stock shifts under Climate Warming (Engelhard et al., 2014; Haug et al., 2017; Mendenhall et al., 2020). At the same time, fisheries will affect future marine fish communities, as they have affected marine fish communities in the North and Barents Seas for decades (Jennings et al., 2004; Johannesen et al., 2012), and current management decisions will contribute to shape fisheries resources in the future, which future modelling approaches are starting to include (Holsman et al., 2020). As such, embracing adaptive management strategies that account for the evolving dynamics of marine ecosystems and fisheries resources is imperative to ensure the sustainability and resilience of our oceans for generations to come.”

- Just an idea, that is probably outside the scope of this manuscript, but to give this manuscript more ecological depth, you could present co-occurrence analyses for all scenarios (see Ovaskainen et al. 2017. Ecology Letters 20:561-576 or the HMSC-book) for all species or just for Arctic seas. It might give some insight in changing co-occurrences (as described in Introduction, L47). I have not yet seen such forecasts with JSDMs, and it might show interesting changing patterns if southerly species arrive in the Arctic.

We agree with the reviewer – in fact, we are already working on this issue in another study. We thought that including species co-occurrences would overload the study with side information not central to our core aim, but since the reviewer 2 also had a similar suggestion, we have added the suggested analysis in the supplementary material in **SM2** and in the text:

Methods lines 236-241:

“Finally, we examined the patterns of species co-occurrences at the level of the spatial random effect. The co-occurrence of specific species is drawn from the covariance structure of the model residuals once the fixed environmental effects have been considered. This analysis reveals pairs of species that either co-occur more frequently or less frequently than expected by random chance, which can partly represent biotic interactions but may also represent patterns arising from spatial patterns associated with environmental covariates not included in the model, or stochastic processes (Norberg et al., 2019; Poggiato et al., 2021).”

Results lines 322-325:

“After accounting for the fixed effects representing species responses to the environment conditioned on their traits, we found pronounced residual species co-occurrence patterns with strong statistical support although ($p < 0.05$, **SM2**).

Clarity and context

The manuscript is well written, concise, with context embedded in results from previous work

References

The authors refer previous literature appropriately.

Reviewer #1 (Remarks on code availability):

The code supporting the analysis is available from the first author's Github repository. This code is annotated, supporting its use by others. I suggested to add a reference to this repository in the main document, allowing easy access.

I did not run it since JSDBMs usually take a long-time to converge. And the data are available from online repositories as cited, meaning that I would have to download these from the provided repositories, etc.

This is now added in lines 241-242: “Code used for this publication is available at <https://github.com/CescGV/JSDBM-Barents-Norwegian-North>.”

Reviewer 2:

This paper presents a compelling investigation into joint species distribution modeling encompassing over 100 fish species within the North Sea/Barents Sea region, projecting outcomes until the end of the century under three distinct emission scenarios. I find the main approach and much of the methodology rigorous and commendable, and the clarity of presentation of the manuscript is noteworthy. However, there are certain methodological concerns that need to be addressed before I can accept the study's future projections and fully endorse the article for publication.

We thank the reviewer for his overall endorsement of the work and the detailed and constructive comments which we believe we have now addressed.

Main issues:

- One of the major issues I have with this research has to do with the different sources of environmental data for model training and future projections. From what I understand these data came from 3 different models, two of them with projections for a past period that were used to train the model (the Copernicus data), and a different model that was used for the future projections under the SSP scenarios. The few times I have attempted to do something like this I have come across compatibility problems; basically, I found the data from different models was often not comparable. If I plotted time series of my variables over a particular region, I often encountered clear discontinuities in the time periods covered by the different data sources. Even using anomalies for the future projections (i.e., adding the difference between the present-day conditions and the future as estimated by a single model to the present-day projections provided by a different model) sometimes still resulted in significant discontinuities, and it was simply not appropriate to combine the different data sources. I would like to see a similar analysis here to convince me that the model data that you are using is compatible. If this is not the case, you may have to limit yourself to using data from the IPSL model only, both for model training and the future projections (also see the next point).

Thank you for raising such an important point in such a constructive way. It is correct that we fitted the model with environmental data from different sources (Copernicus physical forecast and biogeochemical hindcast) than what we used for projections (Institute Pierre Simon Laplace (CMIP6-IPSL) global earth model), but this was not arbitrary. The Copernicus dataset “GLOBAL_ANALYSISFORECAST_PHY_001_024” is an analysis forecast based on real empirical observations, thus providing the most accurate estimations of the environment at the month of sampling, therefore hopefully representing the species’ actual niches. However, because we are not aware that such a forecast exists for biogeochemical variables (in our model phytoplankton and dissolved oxygen), we used the analogous Copernicus layer “GLOBAL_MULTIYEAR_BGC_001_029” for model fitting of these 2 variables.

The dataset used for projections (CMIP6-IPSL) represents an Earth System Model which was parameterized and then used to predict a hindcast (i.e., this model does not directly contain empirical data). We fitted the model with what we believe to be the most accurate data available (Copernicus data) to ensure the most accurate representation of each species’ niche, and we projected the model into the future with the most appropriate global earth system model (CMIP6-IPSL). To use the latter to also fit the model could result in completely incorrect model fits where species niches are not well represented. Another argument to not use global earth system models for model fitting, is that their “historical run” only runs until 2014, and all the available data after that (2015-2022) would need to be either dropped, or fitted using a different run, thus incurring in fitting incongruencies that could bias the final model.

We acknowledge that this is a limitation of the study because future predictions may in fact differ from what will eventuate. That is, the Earth system predictions are uncertain, and there is not complete agreement (as shown with the hindcast) to our best estimates of current conditions. To show the differences between the CMIP6-IPSL “historical run”, and the Copernicus physical forecast, we conducted a correlation analysis using the most relevant predictive variable in our model, which is sea bottom temperature (SBT):

The correlation between both sources of SBT is significant but low across the study area (Pearson correlation = 0.61). However, the discrepancy is mostly in the North Sea, and in the Norwegian and Barents Sea (points North of 60 degrees of latitude) the discrepancy is minor (Pearson correlation = 0.85). This analysis is now **added to the main text** (lines 222-228), and in SM1 (**Extended Figure 6**).

In addition, and comparing the CMIP6-IPSL to the other models (Canadian Earth System Model version 5 (CanESM5), and Norwegian Earth System Model version 2 (NorESM2)) that could potentially be used for the analysis, we show that the CMIP6-IPSL model has the highest correlation with the environmental data used for fitting the model:

Despite this, currently insurmountable, inconsistency, the use of ESM is the only feasible way that we can predict future conditions and estimate possible implications of climate change. However, we can (1) estimate this inconsistency (in our case very low for most of the study area, as shown with the correlation analysis), and (2) re-predict distribution of species' biomass if future improved versions of global earth system models are released, that better represent environmental changes (precisely because we fit the model with current, empirical data representing the species' actual niches). In any case, we have now extended the text advising caution particularly when interpreting projections into future conditions, particularly in the North Sea. Please see lines 222-228 & 486-498.

Lines 222-228:

“To show the discrepancy between the fitting and the predictive datasets (fitting was done with Copernicus datasets, and projections with CMP6-IPSL global earth model), we conducted a correlation analysis between both sources of SBT monthly averages, in the overlapping period of the CMIP6-IPSL historical run, and the SBT data from Copernicus for the coordinates included in this study. This includes all our data between 2004 and 2014. We show that both datasets are significantly correlated, but present discrepancies in the North Sea (Person correlation = 0.61 Pearson correlation excluding North Sea = 0.85) (SM1 Extended figure 6).”

Lines 486-498:

“However, special caution is required when interpreting projections in the North Sea for three reasons: first, because future climate in the region has no analogue anywhere in the model calibration. Because future North Sea conditions are not represented in the study area at present-day conditions, this fraction of the environmental space is poorly sampled during the model calibration. Projections for the less sampled parts of the environmental space are considered less reliable and should be interpreted with greater caution (Owens et al., 2013; Stephenson et al., 2020). Second, the historical run of the global earth system model used to obtain environmental data for future projections show discrepancies in with the fitting environmental data in the North Sea, as the correlation analysis in the supplementary material shows. Finally, in considering richness and dominance estimates, it is highly likely that species expanding their range from outside the study area into the study area concentrate in the North Sea, which is our lowermost region, and these would not be accounted for here. For these reasons, we advise caution in interpreting our results in the southernmost region of this study.”

- I have been trying to find out exactly which Copernicus datasets were used for model training. The reference provided in the article only refers to the “GLOBAL_MULTIYEAR_PHY_001_030” dataset for the physical data, not to the biogeochemical data. However, from the description, I believe the second dataset used is the “GLOBAL_MULTIYEAR_BGC_001_029”. If that is the case, I am concerned that they don’t correspond to the same simulation. The biogeochemical model would have needed to rely on data from a physical model, and ideally, you’d want to use compatible physics and biogeochemistry data that was generated together. From looking at the documentation of both datasets this does not appear to be the case: the data is available at different resolutions, over a different time period, and have been generated using different versions of the nemo model (version 3.1 and 3.6 respectively). I am not fully familiar with Copernicus global datasets, so I am unsure if there is a more suitable alternative or if compatible data just hasn’t been made available yet. Regardless, references for the two individual datasets need to be provided in the text, and if the data is not fully compatible this should be noted. Again, this is not the case for the IPSL model data you used for the future, which would be fully compatible, so the way your model makes use of the environmental data might just not hold with the IPSL dataset and would make me unsure of your future projections. Because of this I would like to see proof that variable correlation was similar for the IPSL dataset and the training dataset.

Thank you for raising this point as well.

We address the correlation analysis in the previous answer, to show high similarity in most of the study area, between the CMIP6-IPSL dataset and the GLOBAL_MULTIYEAR_PHY_001_030 dataset, for SBT.

Regarding the use of two Copernicus datasets that were used for model training, we agree with the reviewer. As acknowledged in the previous answer, we do use a different dataset for fitting the model, and these have different resolutions and come from different versions of the nemo model. We agree that “ideally” it would be best to use compatible physics and biogeochemistry data that was generated together. However, we are not aware of any dataset that includes all environmental variables selected for this study, which is why we use a different dataset for two of the biogeochemistry environmental variables, which are at a lower resolution ($0.25^\circ \times 0.25^\circ$) than the rest of variables ($0.08^\circ \times 0.08^\circ$). We have added the reference of the biogeochemical hindcast in the methods (line 127), and discussed this limitation in the discussion (lines 478-480).

Lines 124-133:

“Sea ice concentration, surface and bottom temperatures, and northward and eastward current components were obtained from the Global Ocean Physics Reanalysis at a resolution of $0.08^\circ \times 0.08^\circ$, while bottom dissolved oxygen, and bottom primary productivity were obtained from the Global Ocean Biogeochemistry Hindcast at a resolution of $0.25^\circ \times 0.25^\circ$, both of which were available through the Marine Copernicus repository (European Union-Copernicus Marine Service, 2023b, 2023a). Bottom depth was obtained from BioOracle at a resolution of 0.08° (Assis et al., 2018). Environmental information was extracted for each sampling point corresponding to the monthly mean of each survey month, except for sea ice, where the annual mean was preferred, because winter sea ice dynamics can highly influence the populations of several Barents Sea marine fishes throughout the year (SM1 Extended Table 1).”

Lines 477-480:

“Similarly, we identified depth and bottom temperature as the most relevant predictive variables in our study, although the relevance of phytoplankton and dissolved oxygen could have been hindered by the lower resolution of these two variables, which were obtained from a different Copernicus dataset than the rest.”

- R2 is used to test model performance of abundance models, and a cutoff of 0.05 is applied as “good enough” model fit. However, explaining only 5% of the variance of the data would seem to me a rather poor fit; is this 0.05 cutoff typically used for abundance SDMs? If so, can you provide references in the text? As it is I remain unconvinced that your abundance results are significant.

We agree that the fitting for several species’ abundances is rather poor, but two things should be considered here. Firstly, the final biomass models are a combination of a two-part model (an occurrence model and a biomass conditional on presence model). The R^2 is only referring to the biomass conditional on presence model within this two-part model. The distribution of the biomass is strongly determined by the occurrence model and shows substantially better predictive performance (not explanatory performance). Secondly, although it is true that threshold of 5% is somehow arbitrary, all models with CV R^2 higher than 0% are informative, but because every fold of CV provides a different value, and most of the average values close to 0% have folds with negative %, we chose a more conservative threshold of 5%. In any case, the values are within what is common in predictive performance of SDMs (see for example Table 3 in: <https://www.researchsquare.com/article/rs-3457413/v1>). This expanded justification is now added in the main text, please see lines 246-254:

“**Biomass models:** From the initial 107 species included in the model, we restricted all biomass-based analyses to species with > 0.05 mean R^2 in the five-fold cross validation biomass model. Although explaining only 5% of the variance of the data may seem a rather poor fit, two things need to be considered. First, the distribution of the biomass is determined by the occurrence model and shows substantially better predictive performance. Second, the threshold of 5% is somehow arbitrary, but not random. All models with CV R^2 higher than 0% are somehow informative, but because every fold of CV provides a different value, and most of the average values close to 0% have some folds with negative %, we chose a more conservative threshold of 5%.”

- I am concerned that the model may be being used to extrapolate far beyond training conditions by considering climate change until the end of the century. This could perhaps be assessed with multivariate environmental similarity surface (MESS) maps to assess where/when the model is predicting in and out of

range. Projections would perhaps need to be limited in time or space to stay within training range. Or if extrapolation is indeed needed, model performance would need to be assessed in a different way to better measure the model's ability to project into novel conditions (for example via block cross-validation rather than random separation of the data).

Thank you for raising this important point. We conducted a MESS analysis using the MESS() function from the modEVA package in R (Barbosa et al., 2013; Elith et al., 2010) (lines 228-236 & extended figure 7). This figure (also copied below this text) show that for most of the study area, and for most of the climate scenarios, MESS values are either close to 0, or positive, revealing that most of the projections occur in well-sampled environments during the fitting process. The exception is the North Sea, where several areas have negative values of MESS, particularly in the southernmost North Sea. This is expected, as the future warming of the North Sea is not analogous to any other region in the current-day conditions of the study area; that is, we do not have data south of this area to act as an analogue for future environmental conditions. We have now added some text discussing these points, lines 487-492.

228-236:

“Moreover, we conducted a multivariate environmental similarity surface (MESS) analysis using the MESS() function from the modEVA package in R (Barbosa et al., 2013), to assess the whether the “environmental space” in our projections, was accordingly sampled during the model training (SM1 Extended figure 7). The “environmental space” is the multidimensional space produced by considering each of the environmental variables as a dimension. Projections in poorly sampled parts of the environmental space are considered less reliable (strongly negative MESS values) and should be interpreted with greater caution (Elith et al., 2010; Owens et al., 2013).”

487-492:

“...because future climate in the region has no analogue anywhere in the model calibration. Because future North Sea conditions are not represented in the study area at present-day conditions, this fraction of the environmental space is poorly sampled during the model calibration. Projections for the less sampled parts of the environmental space are considered less reliable and should be interpreted with greater caution (Owens et al., 2013; Stephenson et al., 2020).”

Some minor comments:

- Models were trained with monthly data (for most variables), yet they are used to carry out projections with annual average data. I don't think this is appropriate, as I would expect different correlations to be observed over different scales, both spatial and temporal. In other words, if you trained your models with annual average data instead, I would expect the correlations would be different and therefore you'd get different models. If the models are trained with monthly data, they need to be used to project with monthly data. If an annual average is preferred, then it should be obtained from the average of the 12 monthly projections, though I think it would be simpler to replace the current annual projections with projections for a particular month. Or alternatively the models could be trained with annual data instead, although this may lower model performance.

Our opinion is that training the model with annual data would worsen the model substantially, because the ecological niche of the species would be estimated at an annual resolution (without considering the monthly variation in environmental conditions). We trained the model with monthly environmental data because this is a scale that species experience and respond to. This gives the likely most realistic representation of the environmental niche. Projecting the fitted model to monthly environmental data and taking the annual mean, or projecting this to the annual monthly mean of environmental conditions results in the same estimates of biomass for a model of a linear nature such as HMSC-JSDM, e.g.,

assuming fish abundance in a certain location is a linear function of monthly sea bottom temperature so that:

$$\text{Abundance} = 15 * \text{SBT} + 3$$

and SBT and monthly abundance values were:

	Jan	Feb	Mar	Apr	May	Jun	Jul	Aug	Sep	Nov	Dec	Annual mean
SBT	2	2	3	4	6	7	7	8	8	6	3	5.09
Abundance	33	33	48	63	93	108	108	123	123	93	48	79.36

then whether we calculate the monthly abundance and average these, or use the annual temperature to calculate the abundance the final prediction is the same:

1. Mean from monthly predicted abundance = **79.36**
2. Mean from annual mean environment = $15 * 5.09 + 3 = 79.36$

- The abstract should specify the period for which projections are made, and the fact that different warming scenarios were considered.

We have added this in the abstract as suggested (Line 22-23).

- At the risk of coming across as pedantic, I would suggest that for this work the terms “project” and “projections” should be preferred over “predict” and “predictions”, which are frequently used in the document. Admittedly many scientists use these interchangeably, but generally in the climate change community there is a distinction made between the two: projections is appropriate when making inferences for the future that are subject to conditions, as in “if A were to take place, then B is likely to happen” whereas predictions are not subject to future conditions but just to the current status of a system, as if when predicting tomorrow’s weather based on today’s conditions. For example, long term future climate data are projections, not predictions, as they consider different “scenarios” for the way society and technology may evolve in the future. But as mentioned many scientists consider these terms equivalent.

Thank you, we agree with this use of terminology, and we have changed the wording throughout the manuscript.

- It would be nice to see the correlation between the species that your model provides, either in the main text or in supplementary material.

We have added the suggested analysis in the supplementary material in SM2 and in the text:

Methods lines 236-241:

“Finally, we examined the patterns of species co-occurrences at the level of the spatial random effect. The co-occurrence of specific species is drawn from the covariance structure of the model residuals once the fixed environmental effects have been considered. This analysis reveals pairs of species that either co-occur more frequently or less frequently than expected by random chance, which can partly represent biotic interactions (Norberg et al., 2019; Poggiato et al., 2021).”

Results lines 322-325:

“After accounting for the fixed effects, representing species responses to the environment conditioned on their traits, we found pronounced residual species co-occurrence patterns with strong statistical support ($p < 0.05$, **SM2**).”

REVIEWERS' COMMENTS

Reviewer #1 (Remarks to the Author):

Dear Authors,

I reviewed the revised manuscript and the rebuttal letter and I am happy with the improvements you made to the manuscript and the answers you provided to my previous remarks. I have no further suggestions or comments.

Reviewer #1 (Remarks on code availability):

Running JSDMs is very time-consuming in most cases. Since the authors offer their rcode in a GIT repository, I assume the code is usable by others and that questions will be addressed if needed by the authors.

Reviewer #2 (Remarks to the Author):

Dear authors,

thanks for addressing my comments and concerns. I believe the manuscript is now substantially stronger as a result, and I am happy to recommend its publication.

A very few small typos/issues I noticed when re-reading your manuscript that you may want to address before publication:

- Lines 136-141: there is a repeated sentence
- Line 217: missing word "for" in "...mean AUC of 0.97 (for) the presence-absence model..."
- Line 230: remove word "the" in "to assess the whether..."
- Line 231: remove the comma in "whether the environmental space in our projections, was accordingly sampled..."
- Extended Figure 6: "Preason correlation" should be "Pearson"
- Extended Figure 7: rather than R's "topo" color palette, would it be possible to use a diverging color palette for the MESS maps, so that it is easy to see where the values are

positive or negative? (for example the "RdBu" scale of package RColorBrewer)

Response to reviewers:

Thank you very much to the two anonymous reviewers for their comments. We have implemented the small typos and changes that reviewer 2 recommended, thank you again, it was a very constructive review process.